# Characterization and Study on Fragmentation Pathways of a Novel Nerve Agent, ‘Novichok (A234)’, in Aqueous Solution by Liquid Chromatography–Tandem Mass Spectrometry

**DOI:** 10.3390/molecules26041059

**Published:** 2021-02-18

**Authors:** Jin Young Lee, Kyoung Chan Lim, Hyun Suk Kim

**Affiliations:** Agency for Defense Development (ADD), P.O. Box 35-5, Yuseong-gu, Daejeon 305-600, Korea; bps178@add.re.kr (K.C.L.); alchemist@add.re.kr (H.S.K.)

**Keywords:** chemical warfare agents, Novichok, degradation, LC–MS/MS

## Abstract

As a first step toward studying the properties of Novichok (ethyl (1-(diethylamino)ethylidene)phosphoramidofluoridate (A234)), we investigated its degradation products and fragmentation pathways in aqueous solution at different pH levels by liquid chromatography–tandem mass spectrometry. A234 was synthesized in our laboratory and characterized by nuclear magnetic resonance spectroscopy. Three sets of aqueous samples were prepared at different pH levels. A stock solution of A234 was prepared in acetonitrile at a concentration of 1 mg/mL and stored at −20 °C until use. Aqueous samples (0.1 mg/mL) were prepared by diluting the stock solution with deionized water. The acidic aqueous sample (pH = 3.5) and basic aqueous sample (pH = 9.4) were prepared using 0.01 M acetic acid and 0.01 M potassium carbonate, respectively. The analysis of the fragmentation patterns and degradation pathways of A234 showed that the same degradation products were formed at all pH levels. However, the hydrolysis rate of A234 was fastest under acidic conditions. In all three conditions, the fragmentation pattern and the major degradation product of A234 were determined. This information will be applicable to studies regarding the decontamination of Novichok and the trace analysis of its degradation products in various environmental matrices.

## 1. Introduction

“Novichok” is the name of a series of nerve agents secretly produced by Russia under the ‘FOLIANT’ program in the latter stages of the Cold War [1]. A Novichok agent, known as ethyl (1-(diethylamino)ethylidene)phosphoramidofluoridate (A234) (Figure 1), was alleged by the British government to have been used to poison Sergei and Yulia Skripal in Salisbury, England, on March 4, 2018, and its identity was confirmed by the Organization for the Prohibition of Chemical Weapons (OPCW). After the British authorities classified this incident as poisoning terrorism based on the use of a novel series of nerve agents, the OPCW ratified all Novichok-based compounds in the Chemical Weapons Convention (CWC) lists in June 2020.

Despite the lack of availability of these compounds for study, there has been speculation as to the nature and effects of the Novichok-based compounds, with reports including a theoretical study of A234 chemistry [2], theoretical medical diagnostics [3], and theoretical toxicological studies [4]. In addition to scheduled chemicals, the detection and identification of non-scheduled chemicals that are characteristic markers (degradation products) of Chemical Warfare Agents (CWAs) also plays a key role in verification analysis for the CWC. The identification of such non-scheduled but specific markers of CWAs aids in deciphering the kind of agent that was present in the sample submitted for off-site analysis. As a consequence, many research groups are developing methodologies for their analyses [5,6,7,8,9,10].

However, to the best of our knowledge, no detailed investigation has been carried out into the fragmentation pathways and degradation products of A234 in aqueous solution. Thus, as a first step toward studying the properties of Novichok, we herein report our investigation into the degradation products and fragmentation pathways of A234 in aqueous solutions of different pH values through the use of liquid chromatography–tandem mass spectrometry (LC–MS/MS).

## 2. Results and Discussion

Initially, the degradation pathway of A234 in a neutral aqueous solution (pH = 7.2) was monitored over time by LC–MS using the fullscan mode. The total ion chromatogram (TIC) and mass spectra for A234 ([M + H]^+^, *m/z* = 225) are shown in Figure 2a. Surprisingly, A234 did not degrade easily under these conditions when compared to the behaviors of G- and V-series nerve agents [11]. As shown in Figure 2b, after approximately 2 months, the peak corresponding to A234 had disappeared completely, and new peaks appeared at RT = 1.09 and 3.19 min in the positive fullscan mode. Moreover, in the negative fullscan mode, a new peak appeared at RT = 1.27 min.

Based on a plausible degradation pathway of A234 and the exact masses of the resulting degradation compounds, ethylhydrogen (1-(diethylamino)ethylidene)phosphoramidate (cpd **1,** RT = 3.19 min), *N,N*-diethylacetimidamide (cpd **2,** RT = 1.09 min), and ethylhydrogen phosphorofluoridate (cpd **3,** RT = 1.27 min) were identified, whereby cpd **1** was the major degradation product in aqueous solution at pH = 7.2 (Scheme 1).

To confirm cpd **1** as the degradation product of A234, its fragmentation pattern was examined by LC–MS/MS using the product–ion scan mode, which can give information regarding fragmentation pathways and positively confirm the unambiguous identification of proposed degradation products. In this case, cpd **1** was eluted at 3.19 min and it was found that its fragmentation process was initiated with a simple charge site rearrangement of the C–N bond (α–cleavage) (Figure 3). More specifically, cpd **1** showed a diagnostic product ion peak at *m/z* 150, which was attributed to the loss of C_4_H_11_N from [M + H]^+^ through α–cleavage. We also studied the fragmentation pattern of A234 by LC–MS/MS product-ion scan mode, and the result is presented in Figure 3b. In contrast to a fragmentation pattern of cpd **1**, A234 exhibited a diagnostic product ion at *m/z* 197, corresponding to [M + H−C_2_H_4_]^+^, which formed through the α–cleavage. These results indicate that the fragmentation patterns of A234 and cpd **1** are similar, but that the diagnostic product ions are different in a neutral solution (pH = 7.2).

We also studied the hydrolysis rate and fragmentation pathway of A234 in a basic aqueous solution (pH = 9.4), whereby the pH of the aqueous solution was adjusted using a 0.01 M potassium carbonate solution. After 24 h, the sample was analyzed by LC–MS/MS in the positive fullscan mode, and it was found that no hydrolysis took place, with only the peak of A234 being observed in the chromatogram. Moreover, after 2 months, A234 remained the major component, although a peak corresponding to cpd **1** was beginning to appear. Based on the above results, it was apparent that the hydrolysis rate of A234 in a basic aqueous solution was significantly slower than in a neutral solution, although the same degradation product was obtained under both sets of conditions (Figure 4, additional data could be found in Appendix A). 

Finally, we examined the hydrolysis rate and degradation products of A234 in an acidic aqueous solution, which was obtained by adjusting the pH of the aqueous solution using a 1.0 M acetic acid solution. In this case, after 3 days, a peak corresponding to cpd **1** was beginning to appear in the chromatogram; however, after 1 week, the peak of A234 had completely disappeared, and peaks corresponding to both cpd **1** and cpd **2** were observed. These results therefore indicated that in an acidic condition, A234 was first easily protonated, causing the protonated intermediate to become more electrophilic, thereby promoting nucleophilic substitution with water [12]. As the result, the hydrolysis rate of A234 is faster than under neutral and alkaline conditions (Figure 5, additional data could be found in Appendix A).

## 3. Materials and Methods

### 3.1. Reagents and Chemicals

All chemicals and reagents required for the microsynthesis of A234 (CAS number 2387496-06-0) were purchased from Sigma-Aldrich (Seoul, Korea). Gradient-grade solvents (methanol and acetonitrile) for LC–MS/MS were purchased from Merck (Seoul, Korea). A234 was synthesized in our laboratory and characterized by nuclear magnetic resonance spectroscopy. The purity of the prepared A234 was >95%.

### 3.2. Liquid Chromatography (LC) Conditions

A Thermo-Scientific Vanquish UHPLC system was for LC analysis, and this system was equipped with a Thermo-Scientific autosampler plus pump and a 100 mm × 2.1 mm Waters CORTECS UPLC C_18_ column (Waters, Seoul, Korea) with a 1.6 µm particle size. The mobile phase consisted of water (solvent A) and acetonitrile (solvent B), each modified with 0.1% formic acid. The gradient was as follows: 1% B (from 0 to 2 min), linear increase up to 99% B at 8 min, hold for 2 min, 1% of B at 12 min, and hold for 3 min. The flow rate was set at 0.2 mL/min. The injection volume for all LC experiments was 10 µL, and this was achieved using an autosampler.

### 3.3. Mass Spectrometer (MS) Conditions

The LC column effluent was introduced into a Orbitrap Fusion Tribrid mass spectrometer equipped with a quadrupole, an Orbitrap, and a linear ion trap (Thermo Fisher Scientific, San Jose, CA, USA), in addition to an atmospheric pressure ionization source/interface operated in electrospray ionization (ESI) mode. The capillary temperature and spray voltage were optimized to obtain a maximum response at *m/z* 225, which corresponds to the [M + H]^+^ ion of A234. The ESI conditions were as follows: spray voltage = 3.5 kV, capillary temperature = 350 °C, vaporizer temperature = 300 °C, sheath gas = 35 arbitrary units, and auxiliary gas = 10 arbitrary units. MS/MS product ion scans were carried out at a collision energy of 25 eV. The collision gas was argon at 1.5 m Torr. Xcalibur software v.3.1 (Thermo Finnigan, San Jose, CA, USA) was used for instrument control, data acquisition, and data handling.

### 3.4. Preparation of Aqueous Samples

Three sets of aqueous samples were prepared at different pH values. A stock solution of A234 was prepared in acetonitrile at a concentration of 1 mg/mL and stored at −20 °C until required for use. The aqueous samples (0.1 mg/mL) were prepared by diluting the stock solution using deionized water (Milli-Q, Seoul, KOREA). The acidic aqueous sample (pH = 3.5) and basic aqueous sample (pH = 9.4) were prepared by adjusting the pH of the original aqueous solution using 0.01 M acetic acid and 0.01 M potassium carbonate solutions.

## 4. Conclusions

In this study, we analyzed and studied the fragmentation patterns and degradation pathways of Novichok, namely ethyl (1-(diethylamino)ethylidene)phosphoramidofluoridate (A234), in aqueous solution at different pH values for the first time. Our results revealed several important details regarding the properties of A234. Firstly, A234 produced the same degradation products regardless of the pH of the aqueous solutions. In addition, the hydrolysis rate of A234 was faster under acidic conditions than under neutral or alkaline conditions. Furthermore, the obtained fragmentation pattern and exact mass data confirmed the identities of the degradation products, whereby different fragmentation patterns and degradation products were discovered upon varying the pH of the aqueous solution. This information will be applicable in studies related to the decontamination of Novichok and the trace analysis of its degradation products in various environmental matrices, such as water, sand, and soil. Currently, we are carrying out further investigations to determine why the hydrolysis rate of A234 is most rapid under acidic conditions, and the results will be presented in due course.

## Data Availability

Data is contained within the communication or Appendix A.

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
