# Peer review of "Characterization and Study on Fragmentation Pathways of a Novel Nerve Agent, ‘Novichok (A234)’, in Aqueous Solution by Liquid Chromatography–Tandem Mass Spectrometry"

_molecules, 2021, doi:10.3390/molecules26041059_

Round 1

Reviewer 1 Report

I have a few comments on the presented article:

  1. In the Introduction, the authors state that Novichok-based compounds have been included in the CWC lists. It should be Schedules of chemicals (specifically Schedule 1). The CWC does not provide cover codes (eg A234), only common names of structural groups and their examples together with CAS. The agent corresponding to A234 has CAS 2387496-06-0.
  2. Figure 1 shows the structure of A234. For comparison, it would be appropriate to list some other Novichok-type agents, or representatives of other groups of nerve agents (G, V, or GV). 
  3. The authors used 0.01 M acetic acid and 0.01 M potassium carbonate to prepare the samples. Why did they choose this particular environment, why not eg HCl, NaOH?
  4. The authors state that the hydrolysis of A234 took place most rapidly in an acidic environment. Conversely, in an alkaline environment (unlike agents G, V) the hydrolysis was slow. This finding is very important from a scientific point of view. Can it be eplained rationally? It seems to me that there is a lack of discussion with literature date (eg reference 2, 4).
  5. Another note on References. I recommend supplementing or replacing citation 1 with more representative publications. I found a typo in citation 7 - the year of publication should be 2015, not 2017.

Reviewer 2 Report

Dear authors of the manuscript “Characterization and study on fragmentation pathways of a novel nerve agent, ‘Novichok(A234)’, in aqueous solution by liquid chromatography-tandem mass spectrometry“,

Your manuscript covers an interesting topic and has a very high potential for publication. However, there are several concerns (listed below) that should be addressed first.

Major concern:

  1. Is it possible to claim (page 3, Results, 1st paragraph, line 4–5) that “A234 did not degrade easily under the conditions when compared with” the classical nerve agents? The measurements you present were performed at cca 0 h when the substance is highly abundant and possibly slightly contaminated, although purity was high. And then, there are results from 2 months when the substance is completely degraded. Results in 1 h or other time intervals (1 day, 1 week) can effectively support your claim. Is it possible to add such results?
  2. Can you support your results regarding acidic conditions with a figure?
  3. There is an inconsistency in the text. Results performed under acidic conditions say that there was “an additional degradation product”, while at the same time, you claim that there were “same degradation products regardless of the pH” in the discussion section (and abstract).

Minor concerns:

  1. Title: add space between “Novichok” and “(A234)”.
  2. Abstract: line 1–2: “as a first step” with “for the first time” sound like too many “first”. I would recommend rewriting the sentence or deleting “for the first time”.
  3. Introduction, line 1–2: “secretly” and “secret”…. I would recommend deleting one of them.
  4. Introduction, line 2–3: “Novichok, which is otherwise known as ethyl….” may give a missing impression that the whole group has just one designation. What about “A novichok agent, known as ethyl….”?
  5. Description of Figure 1: use a lowercase letter in the case of “Ethyl”.
  6. Chapter 2.1, last sentence: “The purity” was “pure” does not make sense. I would recommend rewriting the sentence or deleting one of them.
  7. Chapter 2.2., line 5: delete space between “1” and “%”.
  8. Chapter 2.4, line 4: add the type of machine used for deionized water production (with supplier.
  9. Results, 1st paragraph, page 3, line 5: the comparison with G- and V-series nerve agents should be appropriately quoted.
  10. Page 5, description of Scheme 1: add spaces to “(pH=7.2)” (change it to “(pH = 7.2)“).

I hope my comments will help to improve the manuscript.

Best regards.

Round 2

Reviewer 2 Report

Dear authors of the manuscript “Characterization and study on fragmentation pathways of a novel nerve agent, ‘Novichok (A234)’, in aqueous solution by liquid chromatography-tandem mass spectrometry“,

Your manuscript has been significantly improved. However, I still feel that several relatively minor comments should be addressed.

Semi-major concern:

  1. Is it possible to add all those negative results as supplementary material? I believe it still could enrich the community with useful information. If possible, please, do not forget to check typos in figure descriptions (see minor comments).

Minor comments:

  1. Page 4, figure 4 description: add spaces round “=” in “pH=7.2“.
  2. Page 9, figure 5 description: add spaces in front of all “=” signs.

Best regards.

Author Response

Dear reviewer 2,

Q: Is it possible to add all those negative results as supplementary material? I believe it still could enrich the community with useful information. If possible, please, do not forget to check typos in figure descriptions (see minor comments).

Answer: I added the degradation time profiles for A234 regarding pH conditions as supplementary material. Supplementary figure 1 showed the time profile for A234 in an acidic condition for 1 week. And, supplementary figure 2 showed the time profile for A234 in a basic condition. As shown in the supplementary figure 2, we couldnot find the peak of cpd 3 until 1 week. Unfortunately, I didnot analyze 2 months sample in the nagative fullscan mode because we can find A234 remained the major peak after 2 months. Actually, because the ratio of cpd 1 and cpd 3 is almost 80 : 1, we couldnot find the peak of cpd 3 if not degradation of A234 has gone far enough.

Q: Page 4, figure 4 description: add spaces round “=” in “pH=7.2“.

Answer: Corrected

Q: Page 9, figure 5 description: add spaces in front of all “=” signs.

Answer: Corrected